# Damage Effect of Amorphous Carbon Black Nanoparticle Aggregates on Model Phospholipid Membranes: Surface Charge, Exposure Concentration and Time Dependence

**DOI:** 10.3390/ijerph20042999

**Published:** 2023-02-08

**Authors:** Xiao-Feng Wang, Kun Xu, Xin-Rui Li, Ya-Xin Liu, Jie-Min Cheng

**Affiliations:** College of Geography and Environment, Shandong Normal University, Jinan 250358, China

**Keywords:** model membrane damage, modified carbon black nanoparticle, concentration dependent, electrostatic interaction, giant unilamellar vesicles

## Abstract

Commercial nano-scale carbon blacks (CB) are being harnessed widely and may impose potentially hazardous effects because of their unique properties, especially if they have been modified to grow reactive functional groups on their surface. Cytotoxicity of CB has been well studied but the membrane damage mechanisms and role of surface modification are still open to debate. Negatively and positively charged giant unilamellar vesicles (GUVs) were prepared using three lipids as model cell membranes to examine the mechanistic damage of CB and MCB (modified by acidic potassium permanganate) aggregates. Optical images showed that both anionic CB and MCB disrupted the positively charged but not the negatively charged GUVs. This disruption deteriorated with the rise and extension of exposure concentration and time. Lipids extraction caused by CBNs (CB and MCB together are called CBNs) was found. MCB caused more severe disruption than CB. MCB was enveloped into vesicles through an endocytosis-like process at 120 mg/L. MCB mediated the gelation of GUVs, perhaps through C-O-P bonding bridges. The lower hydrodynamic diameter and more negative charges may have been responsible for the distinction effect of MCB over CB. The adhesion and bonding of CBNs to the membrane were favored by electrostatic interaction and the practical application of CBNs warrants more attention.

## 1. Introduction

Advances in nanoscience and engineering knowledge have breathed new life into industrial production and human life. Nanomaterials are widely used in many fields, such as computer and microelectronics, medicine, biological science, energy production, agriculture, and even environmental control and remediation [1]. Although the broad action of nanotechnology has great implications for societal well-being, more and more scientific and scholarly endeavor was set in motion by questions about its risk and hazardous nature. Beyond doubt, these questions do exist. For example, animal studies manifested that nanoparticles (NPs) can elicit severe lung toxicity [2,3,4] and the first clinical case showed that polyacrylate NPs in the workplace are considered to be the culprit in pulmonary diseases of workers after long-term exposure [5]. Nanomaterials (NMs) may have nano-specific toxicity [6] despite doubts about their “specificity” [7]. They could form nano-bio-interaction interfaces due to their larger specific surface area [8], and generate more reactive oxygen radicals (ROSs, which could induce oxidative damage of tissues) because of quantum size effects [9]. There are also many highly cited articles explaining the mechanisms and influencing factors of the toxicity of material at the nano-level [10,11].

Of all the general mechanisms of toxicity for NMs, cell membrane damage is a significant paradigm. The cell membrane is a barrier between the intracellular structure and the extracellular environment in which exogenous substances are present. After contact with cell membranes, NPs can cause direct physical damage to membranes or indirectly lead to intracellular toxicity through biochemical reactions. Hence, a comprehensive understanding of the interaction mechanism between phospholipid membranes and NPs is the key to explaining the toxicity of NMs. A high number of studies have been dedicated to revealing the biophysical processes of NMs that induce the destruction of phospholipid membrane and transfer across the plasma membrane by various methods, such as cell incubation in vitro [12,13,14], antibacterial testing [15,16,17,18], molecular dynamics simulation on computers [18,19], and a combination of these methods. However, when living cells are used, the dynamic process of interaction at the nano–bio interface cannot be easily observed. Giant unilamellar vesicles (GUVs), composed of closed bilayers and formed by self-assembly of phospholipids, have been well used as substitutes for real cells for targeted studies on membrane damage caused by NMs. It has been suggested that SiO_2_ NPs tend to destroy GUVs due to hydrogen bonding interaction assisted by electrostatic attraction [20]. Atmospheric fine particles adhere to the cationic sites and disrupt the GUV membrane via surface oxidizing groups [21]. As for carbon-based NMs, negatively charged multiwalled carbon nanotubes (MWCNTs) can penetrate and disrupt the GUVs with a positive charge, and extract minor phospholipids from GUVs with a negative charge but not disrupt them. The surface defects, e.g., dangling carbon bonds, exacerbate the interaction of CNTs with the membrane [22].

Carbon black (CB), the industrial form of nano-sized soot, which is a fraction of the black carbon (BC) combustion continuum (the incomplete combustion products of fossil fuels and plants, from char or charcoal to graphite and soot) [23,24], has been widely used as fillers in rubber for tire [25,26], adsorbents to heavy metal in water or soil [27,28,29,30], and other applications. Whether CB is engineered or inadvertently produced in the natural combustion process, it may cause potential harm to humans and organisms in the workplace or general environments, including air, water, and soil. It was been suggested that the bioreactivity of CB is related to its spherulite size and resulting surface area [31]. Installation of 40 μg of ultrafine carbon black increases the percentage of pulmonary neutrophils in mice 24 h later and acid-functionalization enhances their pulmonary toxicity [32]. The toxicity of flakelike-shaped CB on lung tumor cells in vitro is more than MWNTs and carbon nanofibers (CNFs) [33]. However, these studies did not elaborate on the interaction mechanism between CB and biological cells. A previous study showed that CB induced shape change and the bursting of GUVs but did not provide an intuitive image of CB contacting with GUVs [14]. In addition, commercial CB usually required surface acid modification to extend its application in environmental and material science. For example, surface modification (MCB) was used to increase the negative charge of CB and improve its adsorption capacity for heavy metals. It was not entirely clear whether the MCB had potential environmental risks.

In this study, we used two types of GUVs (cationic and anionic) as model cell membranes to mimic the situation of biomembrane exposed to industrial CB and MCB from low to high doses. The purpose of our study was to provide better understanding of the cytotoxicity of CBNs (CB and MCB) from the perspective of the nano–bio interface process and membrane damage.

## 2. Materials and Methods

### 2.1. Materials

Industrial nano-scale carbon black (CB, >95% purity) purchased from a carbon black factory in Jinan, China. Neutral phospholipids DOPC, negatively charged phospholipids DOPG, and a positively charged lipid 16:0 TAP, were kindly provided by Dr. Jiang of Shandong University [22]. The schemes of the molecular structures of lipids are displayed in Figure 1. Sucrose and glucose (analytical reagent) were purchased from Sinopharm Chemical Reagent Co., Ltd. (Shanghai, China). Ultrapure water (Synergy UV, Millipore, Bedford, MA, USA) was used.

### 2.2. Modification, Dispersion and Characterization of CBNs

The CB was further oxidized to obtain modified carbon black (MCB). Briefly, 10 g of CB and 140 mL oxidizing solution containing 20% (*w*/*v*) HNO_3_ and 1.58% (*w*/*v*) KMnO_4_ were mixed at 90 °C for 3 h. The oven-dried CB and MCB powder were diluted to different concentrations with 0.1 M glucose and treated with ultrasound at a power of 100 W for 2 h in 25 °C water bath to shatter large particles, followed by absorbance measure immediately at 800 nm to verify the final concentration. The quantitative curves of CB and MCBsuspension fully dispersed by sodium dodecyl sulfate (SDS) are shown in Figure 2. The final concentration was marked as High (104 mg/L for CB and 120 mg/L for MCB), Middle (67 mg/L for CB and 77 mg/L for MCB, and Low (21 mg/L for CB and 25 mg/L for MCB) in the exposure experiment.

The morphology and particle size of CB powder were observed by transmission electron microscopy (TEM, JEM-2100F, JEOL, Tokyo, Japan), and one hundred particles were counted to calculate the arithmetic mean approximate diameter. The specific surface area (SSA) of the carbons was determined using a Brunauer-Emmet-Teller (BET) multipoint curve method on a 3H-2000PSA4 SSA analyzer (BeiShiDe, Beijing, China). Samples were immersed in the condition of 105 °C for 7 h to remove possible moisture prior to measurement. The characteristics of CBNs suspension (100 mg/L in 0.1 M glucose at pH 6.5) were assessed in triplicate after 2 h sonication. The electrophoretic mobility (U_E_) of CBNs was measured on a Zetasizer Nano analyzer (Malvern Instruments, Malvern, UK) based on Laser Doppler Velocimetry, and the Zeta Potential was calculated simultaneously through Henry Equation based on Smoluchowski estimation (*f(Ka)* = 1.5). Hydrodynamic diameter (D_H_) of the particles was determined using a Zetasizer nano analyzer based on dynamic light scattering (DLS).

### 2.3. Preparation of Giant Unilamellar Vesicles (GUVs)

GUVs with different charges were prepared by gentle hydration method taking advantage of the self-assembly capabilities of phospholipids [22]. Briefly, DOPC, DGPC, and 16:0 TAP were dissolved in a 2:1 chloroform/methanol to obtain a phospholipid stock solution at a concentration of 18 mg/mL, 2 mg/mL, and 2 mg/mL separately. Then, 50 μL of DOPC solution was mixed with 50 μL of DGPC or 50 μL of 16:0 TAP solution in a glass tube to prepare negatively charged GUVs^−^ and positively charged GUVs^+^ respectively. The phospholipid mixture was then blown dry with nitrogen and kept under vacuum for 1 h to remove the residual organic solvent to form a uniform thin layer of phospholipids (1 mg) at the bottom of the glass tube. Then, 4 mL of a 0.1 M sucrose solution was added to the tube, and a GUV stock suspension was formed after hydration at 40 °C for 24 h. GUV suspensions were diluted to 100 mg·L^−1^ with 0.1 M glucose solution before the exposure experiment. Phase contrast between the internal and external of GUVs was enhanced thanks to the stepwise difference in the refractive indices of the sucrose and the glucose media.

### 2.4. Exposure of NMs to GUVs

Briefly, 10 μL aliquots of GUVs^−^ or GUVs^+^ were added to the center of the glass dish and dispersed rapidly with 990 μL of CBNs. Control samples were prepared by diluting 10 μL of GUVs using 0.1 M isosmotic concentration of glucose without CBNs. Then the change of GUVs shape and its interaction with CBNs were observed under an inverted 40× objective microscope in a bright field after a certain time, and images were obtained by a Canon EOS 750D SLR camera. Images were processed to enhance contrast.

## 3. Results and Discussion

### 3.1. Characterization of CBNs

Figure 3 shows the shape and size distribution of CB and MCB particles. It can be seen that the CB particles with rough surfaces had no unity of shape and the average diameter of the individual particles was 47.35 nm, but a heterogeneous particle consisting of a strong aggregation of individual particles was found. Compared to the CB, the particle size (44.15 nm) and morphology of MCB did not change significantly but the size distribution became more uniform. The SSA of MCB was lower than CB by 5% (Table 1). This result is common and is believed to be because the acid modification increases oxygen surface groups at the entrance of the micro-pores, blocking the diffusion of species into them [29,32,34].

The U_E_ and Zeta Potential of CB suspension at pH 6.5 were −0.80 μm·cm/(V·s) and −10.20 mV, respectively. The CB at this time had a slight negative charge or was close to neutral. However, MCB increased its charge density, demonstrated by the values reducing to −3.01 μm·cm/(V·s) and −38.33 mV, respectively. This should be attributed to the addition of oxygen-containing functional groups such as –COO^−^ after acid modification, as described in our previous study [29,35]. FTIR spectra (Appendix A) indicated that compared with the CB, peaks of the FNCB were strengthened and broadened at 3200–3450 cm^−1^ and 1580–1600 cm^−1^, which was attributed to the O–H stretching vibration of the hydroxyl, carboxylic, or adsorbed water molecules and the C=C vibration mode due to the aromatic ring structures, respectively. A new weak peak at 1724 cm^−1^ was assigned to the C=O stretching frequency of the carboxylic groups. These assignments indicated the oxygen-containing functional groups were grafted and enhanced after surface treatments. The D_H_ of CB and MCB were 537 nm and 337 nm respectively. This difference may be attributed to an increase in the surface charge density of the MCB which resulted in particles motioning more separately due to the incrassate electric double layer, and thus strong electrostatic repulsion.

### 3.2. Effect of CB and MCB on GUVs^+^ and GUVs^−^

Figure 4 shows the effect of low-concentration CB and MCB on GUVs with different charges. Only in the GUVs^+^ group, was there a significant interaction between CBNs and the phospholipid membrane. Hardly any disturbance occurred to GUVs^−^, evidenced by the structural integrity even when the exposure time was extended to 12 h.

Specifically, the number of rounded GUVs^+^ decreased in the same field of view after 1 h of exposure to CB dispersion, compared with the control. The vesicles were deformed (black arrow in Figure 4) and some phospholipids were extracted by hydrophobic NPs (white arrow in Figure 4). Such extraction is a typical destructive mechanism when the biomembrane is exposed to graphene nanosheets [18] and MWNTs [22]. For MCB treatment, the profile of GUVs^+^ became non-smooth and the dent or discontinuity of GUVs^+^ was also found. MCB tended to attack a specific “point” of GUVs^+^, but whether this was a cationic point, e.g., N^+^, remains to be proved.

Compared with CB, MCB caused more severe damage to GUVs^+^, as evidenced by more chaos of GUV^+^ bearing MCB (more details below). This difference was presumably due to the stronger charge intensity and the lower D_H_ for MCB than CB (the higher level of aggregation of CB shown in the image supported the result of a larger D_H_ for CB). SSA was a determining factor related to NP toxicity [8] but it did not dominate in this study because MCB had a similar SSA with CB. The electric potential difference between the NPs and the liposomes drove the NPs’ access to the GUVs^+^ surface and their bond to the head group of lipids. D_H_ may determine the way that NPs interact with and transform into the bilayer.

### 3.3. Effect of Concentration Level of CBNs on GUVs^+^ Damage

Since GUVs^−^ were less disturbed by CBNs, the concentration effects of CBNs on cationic vesicles (GUVs^+^) were exploited. According to collision theory, higher concentrations mean more particle activity and the chance of contact with the membrane, in other words, higher toxicity. In low-concentration CBN suspensions, the profile of GUVs was still intact and clear, and only a small number of NPs were adsorbed outside the GUVs after 10 min of exposure (Figure 5, 10 min). In middle-concentration carbon suspensions, more black NPs adhered to the GUVs compared to the low-concentration treatment (Figure 5a(i),b(i)) but pronounced damage did not yet occur. When the concentration continued to rise, a large number of CBNs accumulated around the GUVs. In CB treatment (Figure 5a, High), the GUVs showed a tailed-like protrusion. If a mismatch between the outer and inner leaflets of the bilayers occurs, the GUVs adjust their shape and diminish their curvature to minimize the free energy of the membrane [14,22]. This indicated that the CB attached to the organic skeleton and extracted the lipids molecules from the outer layer (Figure 5a(ii)). More seriously, the extracted phospholipids covered on CB and the little lipids raft were spontaneously wrapped into the GUVs without energy, forming a chaotic state of mixed CB and GUVs (Figure 5a(iii)). Unlike CB, which surrounds vesicles everywhere (Figure 5a(i)), MCB tends to concentrate at a certain point of GUVs^+^ (Figure 5b(i)) and mediate the fusion of two vesicles (Figure 5b(v)). MCB acts as a polydentate ligand with –COO^−^ building a connecting bridge through C–O–P bond and hydrogen bond with ${\mathrm{PO}}_{4}^{-}$ of hydrophilic head group favored by the electrostatic attraction between anionic NPs and N^+^ of lipids, which is not present for CB or for negatively charged lipids (DGPC). Invaginated GUVs and myelin-like phospholipids fragments extracted by MCB were also found. The dent of the membrane may have been related to the transmembrane process of MCB (Figure 5b(iii)), and this phenomenon was not observed in the CB group. The size of the NPs determines their pathways across the membrane [19,36]. The endocytosis analogous pathway of MCB (primary size ~40 nm, D_H_ ~300 nm) probably exists in the process of interaction with the model membrane (25~100 μm) [37]. Although the transmembrane pathway of CBNs is still unclear, scientists have deeply understood the membrane interface processes of other carbon NMs and applied them to the fields of synthetic cells, drug delivery, biosensors, etc. Cryogenic TEM images showed that CNTs (length 5–15 nm, diameter 1.51 nm) almost vertically inserted into 200-nm-diameter DOPC vesicles and spanned both membrane leaflets [38]. A cell uptake model based on TEM data suggested that a single MWCNT entered cells through direct membrane penetration while bundling MWCNTs through endocytosis [12]. Graphene micelles can self-insert inside the phospholipid membrane and form a sandwiched superstructure [39]. These ideas provide an insight into the biomembrane effect of CBNs through the cell uptake of NMs but they do not necessarily signify damage effects [8,10].

Both exposure surface area and concentrations of NPs were reported to correlate with their resultant health effects [40]. Not only direct membrane damage but also airway blocking may occur when the particle concentration increases [41]. Although dose sets in research are much higher than environment-relevant concentrations, essential attention should be paid to the special scenery where high-dose exposure is inevitable.

### 3.4. Effect of Exposure Time on Membrane Disruption

As the exposure time was prolonged, the effects of CBNs on the membrane begin to appear or worsen. In low concentrations, the GUVs exposed for the first 20 min remained intact, but when the exposure time increased to 240 min, the vesicles were filled with CB (Figure 5a(iii)). The incorporated CB-lipid extractives may enter into GUVs through the pores after extraction. For MCB, the fusion of closed GUVs^+^ mediated by MCB formed the gelation of liposomes (Figure 5b(iii)). The gelation of the membrane was also found when GUVs^+^ was exposed to anionic SiO_2_ [20]. The gelation of the membrane was attributed to the hydrogen bonding between the Si-OH and the O-P groups of phospholipid and electrostatic attraction between the negative charge and N^+^ terminus [20]. At middle concentrations, exposure for 20 min resulted in more adherences of CBNs on membranes and more lipid molecules climbed out along incorporated CBNs than for 10 min (Figure 5(ii)). At high concentrations, the disorder of the plasma membrane was exacerbated after 20 min of exposure. Tailed-shaped GUVs appeared after 20 min exposure in low-concentration CB and middle-concentration MCB. In the high-concentration CB group, deformed GUVs^+^ shrank further after 20 min, and this phenomenon was also found in the low-concentration MCB group. Extraction of phospholipids reduced the number of blocks for assembling GUVs and they decreased their volume to adapt to this change. The osmotic effect is an alternative reason for the shape change of GUVs because added CBNs occupy the space of solvent and increase the osmotic pressure in media. The long-time water expulsion from the inner of the GUVs^+^ resulted in a new equilibrium shape for the GUVs. A study displayed the osmotic effect when the GUVs^+^ adjust their shape and even burst when CB is added [14].

The reaction at the nano–bio interface required a certain reaction time to follow the kinetic process. The time-dependent damage effects of CBNs on simulated membranes reminded us of the need to focus on the long-term health effects of organisms after exposure.

### 3.5. Proposed Interaction Mechanisms of CBNs and GUVs

The phospholipids that assemble GUVs consist of a hydrophobic tail and a hydrophilic head with charge. Firstly, the overall hydrophobic CBNs aggregate toward the surface of the overall hydrophobic biomembrane under the electrostatic interactions, hydrophobic interactions, and strong van der Waals attraction from lipids (Figure 5(i)). MCB surface polarization accelerates the process of binding to the hydrophilic head and increases the potential for chemical bond formation. Secondly, biochemical processes such as phospholipid extraction (Figure 5(ii)) and endocytosis (Figure 5(iii)) occur at the nano–biofilm interface. Lipid extraction is an appealing mechanism that has been well applied to graphene nanosheets, as claimed by Tu et al. [18]. Graphene with a narrow edge (thickness ~1 nm) breaks through the obstacles of the head group and inserts into the membrane bilayer facilitated by the dispersion interaction between the *sp^2^* region of graphene and lipids. Once inserted into the phospholipid bilayer, graphene will stay in the hydrophobic region of the phospholipid bilayer to form a hydrophobic track allowing the hydrophobic tail of lipids to climb up along them. The hydrogen bond between H_2_O in media and the head group of lipids is the driving force to overcome the electrostatic interaction (~800 kJ/mol) between the extracted lipid and the piled lipids [18]. For CB and MCB, we infer that the enveloped mode that triggers the extraction is “point-in” due to the surface roughness of the stochastic shape CBNs rather than the edge-in for graphene or pierce-in for CNTs (Figure 5b, Middle and High). In addition to phospholipid extraction, an endocytosis-like process can occur when the adsorption energy is high enough. The difference is that CB is wrapped after enough phospholipids extraction to create large voids (Figure 5a(iii)) while MCB passes directly because of relatively small D_H_ and strong bonding energy (Figure 5b(iii)). NPs partially enveloped into the vesicles can lead to the tearing of the vesicles (Figure 5b(iv)) [42]. Similarly, when the tendency of the vesicles to deform and shrink exceeds the hydrophobic attraction for the arrangement of phospholipid molecules, the vesicles can tear and produce phospholipid fragments [22]. MCB-mediated gelation of GUV^+^ is a critical finding in this study (Figure 5b(v)). MCB draws the neighboring GUVs^+^ close by its adhesive function group “hands” and increases their regional density. Finally, the membrane is damaged by a synergistic effect of lipids extraction, shape shrink, mechanical tearing, and collective gelation when the exposure time and concentration are raised.

### 3.6. Limitations of This Study

First, the single nanoparticle-bio membrane interaction was beyond the scope of this study because the optical microscope was used to observe GUVs (>20 μm in size). CBNs were often aggregated and formed large particles at submicrometer to micrometer level and also could be observed under the optical microscope. Second, the interaction mechanism was mainly based on the hypothesis from microscope images and the properties of nanoparticles. We measured the zeta potentials and FTIR spectra to support our hypothesis. High-resolution techniques were needed to explore interactions at the molecular level. Efforts will be made in fluorescence tracing and TEM observation in the future.

## 4. Conclusions

In this research, anionic CB and MCB disrupted the GUV^+^ (cationic and Zwitterionic lipids) instead of GUV^−^ (anionic lipids), which confirmed that the electrostatic interaction between NPs and lipids played a significant role in the nano-bio interface reaction. The living cell membrane was much more complicated than the model lipid membrane, and contained anionic domains mainly, but also contained relatively scarcer cationic domains, which provided the adhesion sites for negatively-charged CB. The advantage of model membranes is to exclude uncertainties in living cells such as programmed apoptosis and clarify the effects of CB-membrane physiochemical interactions. Our results showed that CB can “nonspecifically” adhere to GUVs^+^ while MCB can contact GUV^+^ at a “specific” point, then they extract lipids from GUVs^+^. CB can be engulfed into GUVs after extraction of lipids while MCB can cause GUVs to dent and be enveloped into GUVs before lipid extraction due to the strong adhesion force. MCB also intermediated the proximity and gelation of GUVs by chemical attraction. Lower D_H_, higher zeta potential, and a richer functional group of MCB can accelerate and exacerbate the process of membrane damage compared with CB, especially in low exposure concentrations. Examining the damaging effect of CB and its modification to the model cell helped provide a better understanding of their cytotoxicity and how they can be prevented from harm.

## Figures and Tables

**Figure 1 ijerph-20-02999-f001:**
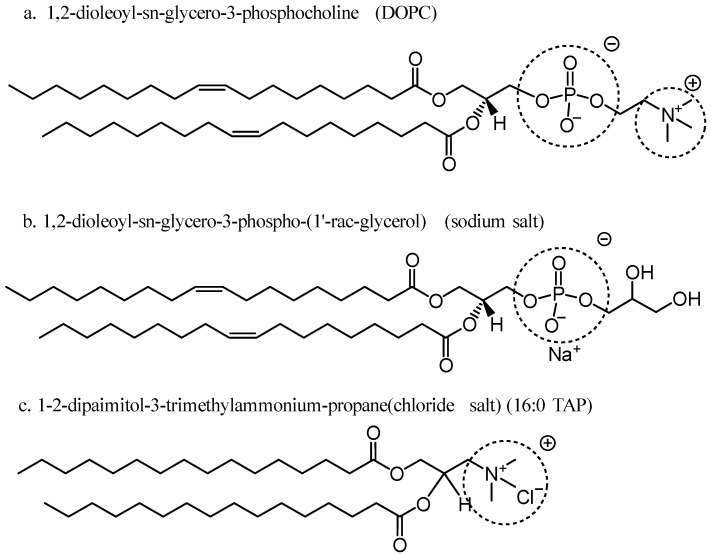
Schemes of the chemical structures of phospholipids with (**a**) zero charge, (**b**) negative charge, and (**c**) a diol-based transfection reagent, positive charge.

**Figure 2 ijerph-20-02999-f002:**
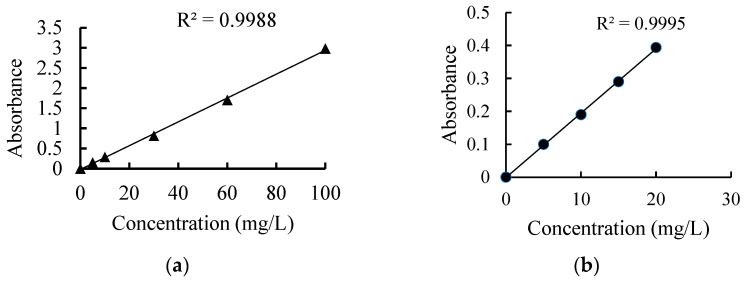
The quantitative standard curves of CB (**a**) and MCB (**b**) suspension using SDS as dispersant by measuring its absorbance at 800 nm.

**Figure 3 ijerph-20-02999-f003:**
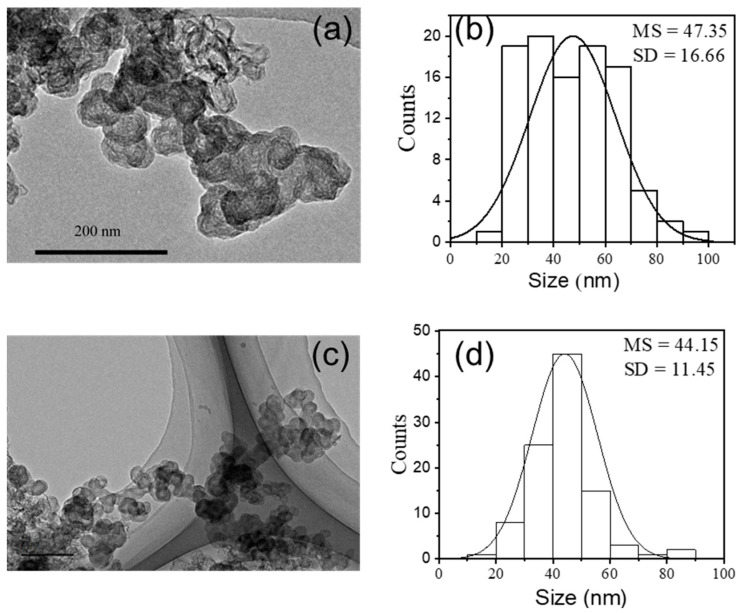
TEM images (**a**,**c**), and histogram and Gaussian fit of the size distribution (**b**,**d**) of CB (**a**,**b**) and MCB (**c**,**d**) particles. MS indicates the mean size and SD means the standard deviation (n = 100).

**Figure 4 ijerph-20-02999-f004:**
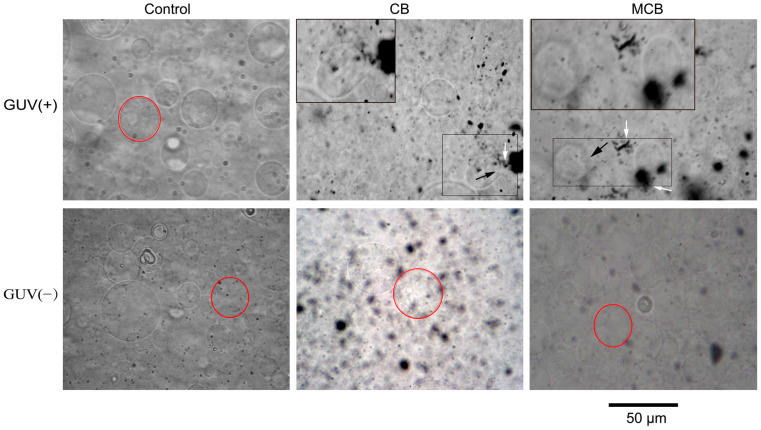
Microscopic images of GUV^+^ exposed for 1 h and GUV^−^ exposed for 12 h in low-concentration CB and MCB suspension. Inserts, magnified views of boxed areas. Red false color lines added to highlight the intact lipid bilayer. Black arrow indicates the deformed GUV. White arrow indicates the phospholipid fragments extracted by NPs.

**Figure 5 ijerph-20-02999-f005:**
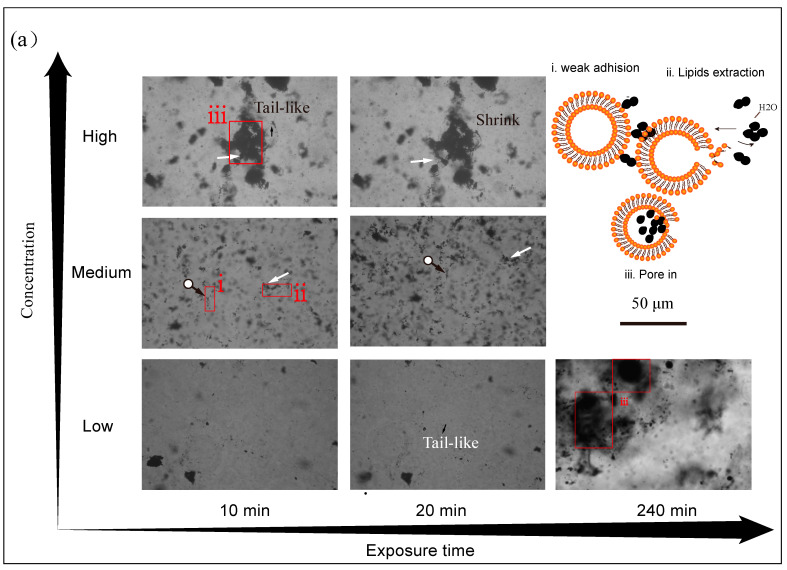
Morphological changes of GUV^+^ at different exposure times in different concentrations of CB (**a**) and MCB (**b**). The black arrow indicates the deformed GUV. White arrows mark phospholipid fragments extracted by nanoparticles. Circled arrows indicate nanoparticles covering the vesicles. Right of (**a**) from i to iii and down of (**b**) from i to vi are modeling selective box areas marked with i to vi (orange beads and brown lines: head groups and tails of lipids; black ball: nanoparticles). High (104 mg/L for CB and 120 mg/L for MCB), Middle (67 mg/L for CB and 77 mg/L for MCB), and Low (21 mg/L for CB and 25 mg/L for MCB) in the exposure experiment.

**Table 1 ijerph-20-02999-t001:** Physiochemical properties of CBN powder and suspension.

CBN	pH	Primary Size (nm)	SSA (m^2^/g)	D_H_ (nm)	U_E_ (μm·cm/(V·s))	Zeta Potential (mV)
CB	6.5	47.35	635.96	537 ± 28	−0.80 ± 0.09	−10.20 ± 1.22
MCB	6.5	44.15	603.38	337 ± 13	−3.01 ± 0.33	−38.33 ± 4.22

## Data Availability

The data presented in this study are available on request from the corresponding author.

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
