# Peer review of "Damage Effect of Amorphous Carbon Black Nanoparticle Aggregates on Model Phospholipid Membranes: Surface Charge, Exposure Concentration and Time Dependence"

_ijerph, 2023, doi:10.3390/ijerph20042999_

Round 1
Reviewer 1 Report
CB can absorb many metals/pollutants from the atmosphere,
did your study account for the purity of CB.
Reviewer 2 Report
In this manuscript, the authors prepared two giant unilamellar vesicles with different type of charges on the surface. The vesicles were constructed by either positively charged or negatively charged commercial phospholipids. Destruction of positively charged vesicles was then observed by authors under an optical microscope when treating the colloid solution with industrial carbon black or modified carbon black. There are serious questions regarding to the actual modified carbon black structures used in this study as well as the exact mechanism the carbon black undergoes to damage the giant unilamellar vesicles, with some comments as follows.
1. What are the dominant functional groups on the modified carbon black to be involved in the vesicle destruction? Different functional groups on carbon black surface can be generated by different oxidation process such as chemical treatment and ultrasonication. Without knowing the exact functional groups participated in the vesicle destruction, mechanistic damaging effect interpretation of experimental results may be difficult and challenge. Characterization of the modified carbon black is needed to make proper discussions and conclusions.
2. Vesicle destruction process in microenvironment (at least on nanometer level) cannot be observed from optical microscope. Especially such process involves interactions at molecular level, such as charge interaction, Van der Waals force, hydrogen boding and even chemical reactions. Authors proposed possible mechanisms based on other people’s work and hypothesis without actual supporting experimental data. Research articles cited by authors are mainly from the results of graphene and carbon nano tubes. Graphene, graphene oxide, carbon black and carbon nano tubes are completely different carbon nanomaterials and will have huge differences in biological system. Hypothesis of the mechanism should be very comprehensive and conservative. But again, discussions and conclusions of research articles should not solely rely on hypothesis.
3. Microscope images and illustrations in Fig 4 and Fig 5 have very low resolution, details and words are too blurry to read. Discussions based on these figures cannot be correlated clearly.
4. Both carbon blacks under optical microscope look like large aggregates rather than carbon black nanoparticles at nanometer level. Vesicle destruction caused by such aggregates is inconsistent with the article title which should mainly focus on the “damage effect of amorphous carbon black nanoparticles on model phospholipid membranes”
5. The chemical 16:0 TAP (Fig. 1c) is not a phospholipid by chemical definition. This compound did not seem to be involved in any way through the paper.
6. Description in line 107-114 is very confusing. If not a mixture of CB and MCB was used for absorbance measurement, should the quantitative curve of MCB be also included?
7. Should the TEM image of MCB be also included?
8. A number of typos and errors needed to be corrected as follow: (a) typo in line 13; (b) inconsistent font size in line 70 – 71; (c) proper noun should be followed by abbreviations in line 12; (d) please define all the abbreviations when they first appear in the paper, such as CNBs and CBNs in line 13 – 16; GO in line 184 – 187 etc.
9. There are many grammar and linguistic mistakes throughout the paper. To prevent confusion, and to provide better understanding for readers, major level English language proficiency should be thoroughly checked.
Reviewer 3 Report
This manuscript reports the damage effect of carbon black on phospholipid membranes, which is directly related to the safety of nanomaterials. Based on the observation of morphological development for GUVs, the damage mechanism was ascribed to the lipids extraction of carbon black and the endocytosis of phospholipid membrane. I think this manuscript might be publishable if the following concerns are well addressed.
1. The microscopic images listed in Figs. 4&5 should be improved.
2. Ref. 1 was published in 2011. I wonder if the estimated nano product market for 2020 was realized or not.
3. The abbreviation of some chemicals is confused, such as DGPC vs DOPG, TAP vs TPA ...
4. In the bottom line of page 5, what does "GO" stand for?
5. In line with Fig. 3b and Table 1, the primary size of CBNs is 47.35 nm. However, it is stated in line 157 that the average diameter of the individual particles is 43 nm.
6. Detailed effect of surface acid-modification on the damage of phospholipid membranes need to be clarified.
7, English should be improved.
Line 10 : negative and positively charged
Line 106: KMNO4
Line 137: negatively charged GUVs+
......
Reviewer 4 Report
Review comments:
The manuscript “Damage Effect of Amorphous Carbon Black Nanoparticles on Model Phospholipid Membranes with Opposite Charge: Exposure concentration and time dependence” mainly reported the mechanism of carbon black damage cell membrane with opposite charge. A possible interaction between control membrane and nano carbon black with opposite charge was studied. And a possible mechanism was proposed. This reported effect introduces a possible damage from environmental CBs to cell. I think this paper can be accepted after a major revision:
(1) In manuscript, figures with higher DPI should be uploaded. It is too difficult to tell the details of the images. Recheck plz.
(2) I recommend that the title could focus more on the surface acid-functional group modification which would help readers get the highlight.
(3) Page 1 line 13 & 16, provide the definition of “CNBs” & “CBNs” plz.
(4) Page 2, line 58-60. A more detailed reason of choosing GUVs as a substitute of a living cell should be provided. And I just wonder whether GUVs could works the same as the living cell, could you provide evidence of the result that living cell could performs the same under the same situation to prove that GUVs could works the same.
(5) In figure 3, the histogram and Gaussian fit of the size distribution between 40-60 nm is apparently lower than size around 20-40 nm and 60-70 nm, please provide more evidences.
(6) Page 5, figure 3. A TEM image of MCB should be provided.
(7) Page 8, figure 5(a)&(b), I wonder if author could quantify the concentration of CB&MCB?
(8) The reason why set a surface acid-functional group modification-CB as control should be provided. And author mentioned in line 340-341 that living cell membrane possess negatively charged, a more convincing reason could help readers to understand the manuscript.
(9) The contribution of this study to real living cell should be supplied in conclusion part.
Reviewer 5 Report
It is found different font size of writing. They should be the same size of fonts. For example, page 2 line 70 word 'fraction out' is bigger than others.
It is not clearly stated the method in the abstract.
Also, it is not clearly stated the results in the abstract.
The resolution of Fig. 3, fig. 5 should be increased
Round 2
Reviewer 2 Report
Authors' response to the comments is very much appreciated. However, the revised manuscript itself did not reflect answers to all the issues and questions in the original comments, even though those issues have been addressed properly by authors. Please see attachment for details.

Reviewer 3 Report
All of my concerns are well addressed.
Author Response
Thank you for your affirmation.